# Supercooling of the A phase of ³He

Y. Tian[1], D. Lotnyk [1], A. Eyal[1,2], K. Zhang[3,4], N. Zhelev [1,5], T. S. Abhilash [1,6], A. Chavez[1], E. N. Smith[1], M. Hindmarsh[3,4], J. Saunders[7], E. Mueller [1] & J. M. Parpia [1] ✉

Because of the extreme purity, lack of disorder, and complex order parameter, the first-order superfluid ³He A–B transition is the leading model system for first order transitions in the early universe. Here we report on the path dependence of the supercooling of the A phase over a wide range of pressures below 29.3 bar at nearly zero magnetic field. The A phase can be cooled significantly below the thermodynamic A–B transition temperature. While the extent of supercooling is highly reproducible, it depends strongly upon the cooling trajectory: The metastability of the A phase is enhanced by transiting through regions where the A phase is more stable. We provide evidence that some of the additional supercooling is due to the elimination of B phase nucleation precursors formed upon passage through the superfluid transition. A greater understanding of the physics is essential before ³He can be exploited to model transitions in the early universe.

The condensation of ³He pairs into a superfluid state occurs via a second-order phase transition at a pressure-dependent transition temperature, $T_c$, shown in Fig. 1. The anisotropic A phase is favored at high temperatures and pressures, while the isotropic B phase is the stable phase below the $T_{AB}(P)$ line[1,2]. In zero magnetic fields, the equilibrium phase diagram exhibits a polycritical point[3] (PCP) at which the line of first-order transitions ($T_{AB}$) intersects the line of second-order transitions ($T_c$) at 21.22 bar and 2.273 mK. The transition between the A and B phases is first order and thus subject to hysteresis. At the PCP, the bulk free energies of the A, B superfluid phases and the normal state are equal.

The A phase is highly metastable, and can persist down to extremely low temperatures for long times (≥1 day) at high pressures, providing surfaces of the container are smooth[4,5]. Standard homogeneous nucleation theory[6,7] argues that the transition from metastable A to stable B is mediated by thermal fluctuations that produce bubbles of characteristic size $r$. For small bubbles (size less than the critical radius, $R_{crit}$), the interfacial energy cost ($\propto r^2$) is larger than the bulk free energy gain ($\propto r^3$), but for large bubbles the opposite holds. Thus, if thermal fluctuations create a bubble with $r < R_{crit}$, it rapidly shrinks. Conversely, a bubble with $r > R_{crit}$ will grow. This model[8],

applied to ³He, leads to $R_{crit} \approx 1.5\,\mu m$, and an activation energy that is many orders of magnitude above the thermal energy[9–11] implying an unobservably small nucleation rate. Surface defects potentially alter the energetics (most surfaces favor the A phase[12] and there is no clean explanation of how they would mediate the A–B transition). Despite extensive experimental[4,13–20] (Fig. 1) and theoretical investigations[21–26], the mechanism for B phase nucleation remains a mystery.

Laboratory studies of the dynamics of first-order phase transitions have cosmological implications, as the statistical theories of the decay of a metastable state in condensed matter[7] are non-relativistic analogs of the quantum field theories used in cosmological models[27,28]. Importantly, the possibility of a first-order electroweak symmetry-breaking phase transition[29,30] in the early universe has been used to explain baryon asymmetry[31]. The same physics also produces gravitational waves[32–35] whose detection are science targets for future space-based detectors such as Laser Interferometer Space Antenna (LISA)[36,37]. Experimental confirmation of the applicability of this model of first-order phase transitions to a laboratory system (whether in ³He or in cold atom systems[38,39]) would lend more weight to the calculations of gravitational wave production for LISA and other future probes of the early Universe. However, the theory of first-order phase transitions in

[1]Department of Physics, Cornell University, Ithaca, NY 14853, USA. [2]Physics Department, Technion, Haifa, Israel. [3]Department of Physics and Astronomy, University of Sussex, Falmer, Brighton BN1 9QH, UK. [4]Department of Physics and Helsinki Institute of Physics, University of Helsinki, PL 64, FI-00014 Helsinki, Finland. [5]Center for Applied Physics and Superconducting Technologies, Department of Physics and Astronomy, Northwestern University, Evanston, IL 60208, USA. [6]VTT Technical Research Centre of Finland Ltd, Espoo, Finland. [7]Department of Physics, Royal Holloway University of London, Egham, TW20 0EX Surrey, UK. ✉e-mail: jmp9@cornell.edu

the early Universe[27] is based on the same homogeneous nucleation theory which fails to explain the behavior of ³He in contact with "ordinary surfaces"; if there is a much more rapid intrinsic nucleation mechanism in operation, the gravitational wave signal could be rendered negligible[35].

Here we study the nucleation of B phase in a pair of chambers connected by a high aspect-ratio "letterbox" channel (Fig. 2). Both the geometry and the surface qualities are relevant: The smaller chamber

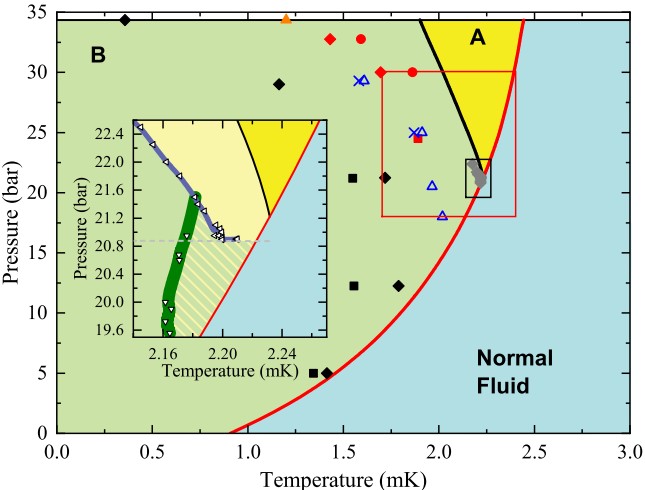

**Fig. 1 | Previous supercooling results.** The equilibrium phase diagram for superfluid ³He. The normal fluid (blue), stable A phase (yellow) and the B phase (green) are separated by a red line that marks $T_c$. The equilibrium A–B transition in zero magnetic field (black line) terminates at the polycritical point (PCP) where the A, B and Normal phases coincide. Centered on the PCP is the region investigated in ref. 20 (black rectangle) and inset where left-pointing triangles show supercooling extent (light yellow) under constant pressure, and downward pointing triangles, pressure decreased conditions (hatched yellow-green region), all in ≤0.1 mT. The region investigated in this paper is shown as a red box. Results from previous investigations in a variety of magnetic fields are shown as gray diamonds: 4.9 mT, 0.5 mT[13]; blue crosses: 56.9 mT, blue triangles: 28.4 mT[14]; orange triangle: 0 mT[57]; red circles: 0 mT, red diamonds: 10.0 mT, red square: 20.0 mT[58]; black squares, black diamonds 28.2 mT[5].

(denoted the Isolated Chamber (IC)) incorporates "ordinary" as-machined coin silver surfaces. It also houses a quartz fork whose resonant properties (frequency and quality factor, $Q$) allow us to infer the phase of the ³He in the chamber. The IC is separated from a larger chamber containing sintered silver by a micromachined channel construction consisting of a 1.1 μm tall × 3 mm wide × 100 μm long channel and 200 μm tall × 3 mm wide × 2.5 mm long lead-in channels on either side. This construction was nanofabricated in silicon and capped with glass[40] (see also Supplementary Note 1). The silver sinter-containing chamber (denoted the Heat Exchange Chamber (HEC)) incorporates a quartz fork similar to that contained in the IC. The A phase is stabilized in confined spaces, and the narrow channel potentially prevents the propagation of an A–B phase boundary from one chamber to the other −allowing the transitions to be independent.

In an earlier publication[20], we reported initial observations of the reproducibility of B phase nucleation and an unexpected path dependence for the A phase's stability. From those experiments, it was unclear whether the path dependence was limited to the region near the PCP and there were few clues about the microscopic origin of the phenomenon. Here, we have expanded the region investigated to include the highest pressure readily accessible to us (the pressure of the minimum of the ³He melting curve, 29.3 bar) and have designed a series of protocols that provide significantly more clarity about the phenomenon.

As already emphasized, homogeneous nucleation theory is unable to explain the nucleation of the B phase from the A phase: There is a vanishing probability that thermal fluctuations produce a B phase bubble that is larger than the critical radius. The transition can be triggered[4,17,41] by bringing a radioactive source near ³He−which is consistent with models where energetic particles (either deliberately introduced or due to Cosmic rays) are responsible for the observed A−B phase transition[11,17,41]. Those models cannot explain why when ³He is repeatedly cooled[14,20], the transition consistently occurs along the same temperature and pressure line (dubbed the catastrophe line[14]). Alternative explanations have been sought. The theory of quantum tunneling of metastable states in field theory[42] has been applied to the ³He system[10], without substantially changing the mismatch in rates between theory and experiment. More complex field-theory-based models such as Q balls[25] or Resonant Tunneling (RT)[26] have been proposed; they are, however, not consistent with the path dependence

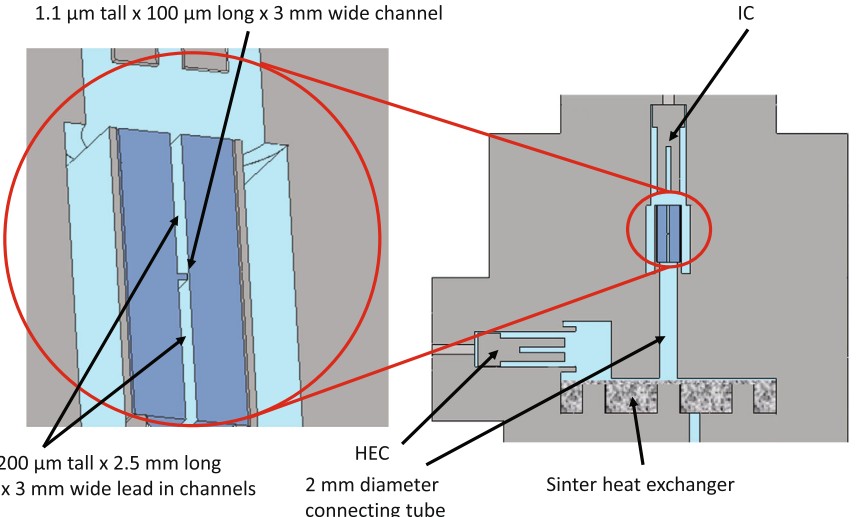

**Fig. 2 | Schematic of cell.** The isolated chamber (IC) and heat exchange chamber (HEC) both contain quartz forks whose quality factor is monitored to determine the phase of the ³He superfluid. The chambers are separated by a 100 μm long channel with aperture 1.1 μm tall and 3 mm wide (the 3 mm width is hidden in the main view). As seen in the enlarged view (circled in red), this "letterbox" channel has two 200 μm tall 3 mm wide 2.5 mm long channels on either side, one opening to the IC, the second connecting to a 2-mm diameter cylindrical tube opening into the HEC.

seen in our experiments. The presence of topological defects such as vortices can enhance the nucleation rate[21]. Such vortices can be detected via calorimetry[17], but our temperature measurements are insufficiently precise. Nonetheless, in our experiment, we do not expect to have a substantial number of vortices: Vortices are either shed during fluid flow or produced during rapid cooling[43,44]. Our flow and cooling rates are very low. In our previous experiment[20], we found that the degree of supercooling was independent of the rate at which we passed through $T_c$—indicating that Kibble–Zurek vortices are not relevant. Moreover, vortex-induced enhancement of the nucleation rate is expected to be too small to explain our observations[21].

In an attempt to explain our observations, we note that the silver sinter contains a large number of chambers that are connected to the bulk fluid by narrow channels or constrictions. We hypothesize that, upon traversing $T_c$, the A phase is formed in bulk, but regions of distorted order parameters are formed in some of these chambers. They act as precursor "seeds" of the B phase. For computational expediency, we treat these chambers as if they are filled with B phase, and refer to them as "B phase seeds". However, confinement would result in a distorted order parameter quite different from the bulk B phase. Surface tension stabilizes the requisite A–B domain walls at sufficiently small constrictions. The size of the largest stable domain wall depends on pressure and temperature: In the A region of the phase diagram, the A phase will rush into any of the chambers whose opening is larger than this size. Conversely, the (path-dependent) catastrophe line will be determined by the size of the smallest constriction that connects to a B-filled chamber. This model is similar to the *lobster-pot* scenario which was proposed for understanding the nucleation of the A phase from B[23].

Cavities in the sinter are unable to explain all of our observations, and it is likely that some other mechanism is also at play. For example, the A phase order parameter (in standard experimental geometries) may contain complicated textures with highly frustrated points that may act as seeds for the B phase. Such seeds may involve B-inclusions, or just precursor regions where the A order parameter is strongly suppressed. While some of this structure forms spontaneously due to the Kibble–Zurek mechanism[43,44], much of the spatial complexity is likely due to surface effects: surfaces constrain the components of the order parameter[45] and surface corrugations or scratches can lead to complicated disgyrations and other structures[46], perhaps containing precursor seeds of B phase. Similar to the cavity scenario, the observed A–B transition is set by the size of the "largest" seed, whose catastrophe temperature is highest. These largest seeds are also the most fragile and may be eliminated by exposure to high pressures where the A phase is most stable. The key feature of the path-dependence in both scenarios is which seeds survive. Development of an understanding of this pressure dependence is essential if $^3$He is to be a useful model for phase transitions in the early Universe.

We emphasize that the order parameter of helium is contained in a high dimensional space, and the paths connecting the A and B phases are strongly influenced by surfaces, textures, and distortions from confinement. Nucleation can occur through both thermal fluctuations and quantum tunneling, the latter of which can display interference effects that are particularly sensitive to such changes in the energy landscape[42]. Models of nucleation in inhomogeneous settings contain a multitude of complexities[47].

## Results
### Experimental details
The normal-superfluid and A–B transitions were detected using quartz forks located in the IC and HEC. The temperatures were obtained with reference to a $^3$He melting curve thermometer[48] mounted on the cold plate of the nuclear demagnetization stage. For details of the operation of the forks and of the thermometry, we refer to the Methods section.

### Supercooling at constant pressure
The first set of measurements was carried out while cooling at a constant rate ($\leq 10\,\mu K/h$) and fixed pressure. Figure 3a shows the temperatures at which the A–B phase transition was detected in the HEC (pink triangles) and the IC (blue triangles). Below 23.8 bar, the HEC transition occurs at a substantially higher temperature than in the IC. In this regime, we believe that the A phase is stable in the channel: It acts as a plug, preventing the A–B wave-front from propagating from the HEC to the IC. The silver sinter in the HEC leads to more complicated variations of the order parameter, and it is reasonable that the HEC and IC contain different B-phase seeds with different catastrophe temperatures. Between 23.8 bar and 26 bar, there is a decrease in the separation between the two transitions, which suggests that the A–B wave-front is only weakly pinned by the channel. Above 26 bar, the two transitions happen simultaneously, and we conclude that in this regime, the channel is unable to sustain an A–B interface once the transition is initiated in the HEC (see the Discussion section and Supplementary Notes 2 and 3).

### Supercooling after decreasing pressure
In our model, the largest degree of supercooling should occur for trajectories passing through the regions where the A phase is most stable (i.e., at high pressure). To explore this feature, we first cool at our highest accessible pressure (29.3 bar) followed by depressurizing and further cooling. In Fig. 3b we illustrate several such trajectories. The solid black lines show cooling trajectories where we maintained an approximately constant $Q$ of the fork in the IC. This constant-$Q$ condition yields a path that is roughly parallel to $T_c$. For these trajectories, we depressurized by 4-6 bar during the first day, followed by proceeding at 1.3 bar per day, or less. During the rapid part of the ramp, but not during the slow part, there was some viscous heating observed in the IC.

We found that the extent of supercooling was significantly greater than what we achieved while cooling at constant pressure (denoted by pink (HEC) and blue (IC) lines instead of data points in Fig. 3b). With the exception of the four lowest constant $Q$ runs (closest to $T_c$), the A–B transitions occurred simultaneously in both chambers, and are depicted in Fig. 3b as coincident crosses and squares. The same symbols (crosses for IC and squares for HEC) are used to denote the observed $T, P$ coordinates of the pressure-varied transitions for the four lowest points. A temperature correction is applied to the IC data to account for thermal offsets between the chambers.

To further explore the path dependence, we considered the trajectories shown as dashed lines in Fig. 3b. These begin with constant-pressure cooling at 29.3 bar, followed by fixed temperature depressurizations and fixed pressure cooling. In all cases, we observe significantly more supercooling than in Fig. 3a. Crucially, there appears to be a definitive locus of points in the $T–P$ plane on which all of the trajectories fall. As illustrated by the dotted black and cyan paths (terminating at 25 and 27 bar), one finds the same A–B transition points when cooling after depressurizing or depressurizing at a constant temperature – as long as the trajectory passed through the A phase at 29.3 bar.

In Supplementary Note 4 we present a detailed comparison between the 23 bar fixed pressure run, and one of the trajectories passing through 29.3 bar before cooling at 23 bar. We find that the only detectable difference is the temperature of the A–B transition. There are no signs of thermal gradients, viscous heating, or other systematic effects.

### Other supercooling results
Figure 4 illustrates four additional runs, each of which involves cooling at 23 bar. The blue curve shows the quality factor of the quartz oscillator in the HEC during cooling. It jumps discontinuously at $T = 2.12\,mK$ ($T - T_c = -0.113\,mK$), indicating the A–B phase transition. For the other

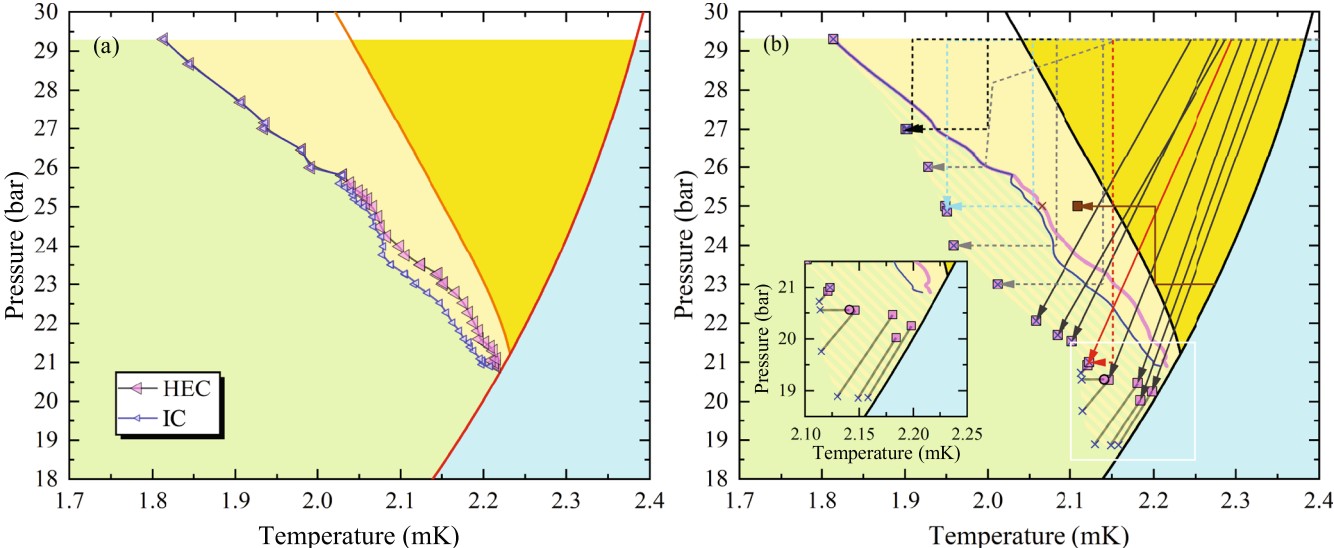

**Fig. 3 | Constant-pressure and pressure-varied A–B transitions. a** Constant pressure-cooled A–B transitions are shown with pink left-pointing symbols denoting transitions in the HEC, blue open left-pointing triangles denoting transitions in the IC. Above 25.8 bar, the transitions are coincident in time and are shown as nested triangles. Color coding follows that in Fig. 1. **b** A–B transitions observed while decreasing the pressure starting from 29.3 bar are shown along with their paths. Pink squares show transitions in the HEC, blue crosses show transitions in the IC. Where these transitions occur simultaneously, they are superposed. At low pressure, they separate with the A–B transition in the IC observed at a lower $P$, $T$ than that in the HEC (see inset). Constant pressure-cooled transitions from panel (**a**) are shown as solid lines. Cyan, black and red lines each show two different paths terminating at the same ($P$, $T$) coordinates. The brown line shows the result of pressurization followed by further cooling at constant pressure. Hatched region marks enhanced path dependent supercooling.

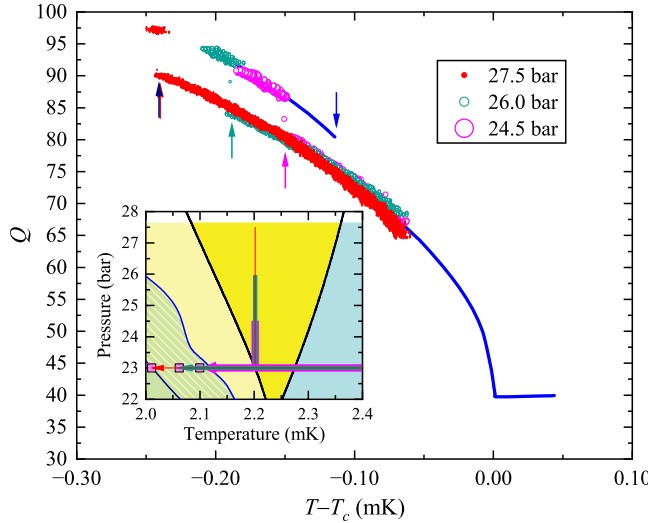

**Fig. 4 | Comparison of constant-pressure and pressure-cycled runs—$Q_{HEC}$ vs$T$ cooled through $T_c$ and $T_{AB}$ at 23 bar.** The constant pressure-cooled experiment in the HEC is shown as a solid blue line. For each of the three pressure-cycled runs, after cooling through $T_c$, while the temperature was maintained at ≈2.2 mK, the ³He was pressurized to 24.5 bar (purple), 26 bar (green) and 27.5 bar (pink), then depressurized to 23 bar, and then cooled further at constant pressure till the A–B transition was observed. The HEC and IC transitions were simultaneous for the 26 and 27.5 bar runs. Arrows mark the positions of the various A–B transitions. The inset shows the A–B transitions and the paths in the $P$, $T$ diagram. The hatched region in the inset is the same as in Fig. 3b.

three runs, the helium is cooled to 2.2 mK, and then slowly pressurized to $p_{max}$ = 24.5 bar, 25 bar, or 27.5 bar. The pressure is then reduced back to 23 bar, and the temperature is reduced further. As expected from our model, the degree of supercooling is a monotonic function of $p_{max}$: the B phase seeds are suppressed by excursions deep into the equilibrium A phase. In these regions the free energy differences

between the A and B phases are largest compared to the thermodynamic barriers. Note, the changes caused by these excursions are subtle enough that they do not appreciably change the quality factors (aside from shifting the A–B transition).

The brown path in Fig. 3b illustrates the reverse effect. We traverse the stable A phase at 23 bar. We then increase the pressure to 26 bar before continuing to cool. We find that the A–B transition occurs at a higher temperature than if we simply cool at 26 bar. This path avoids the regions of the phase diagram where the A phase is most stable.

To ensure that the supercooling is not significantly affected by the sweep rate, we repeated the experiment in Fig. 4, varying the rate of pressurizing and depressurizing during the jog from 23 bar to 27 bar and back. We varied this rate from 1.3 to 27.5 bar/day, finding no difference in the degree of supercooling after completing the cooling at 10 μK/h.

To verify the stability of the A phase obtained after depressurization, we selected a trajectory that terminated below the PCP from Fig. 3b. After cooling through $T_c$, starting from 29.3 bar and 2.15 mK, we depressurized (at fixed $Q$) to 20.5 bar and stopped at a point within 3 μK of the temperature where we previously observed the transition. We waited at this $T$, $P$ for 1 day. We then slowly cooled at a rate of 0.5 μK/h until we observed the transition in the HEC approximately 2 μK below the previously observed result (open pink circle in Fig. 3b). Thus the supercooled A phase has a lifetime in excess of 24 h, and any dynamics which happen on this timescale do not appear to significantly influence the catastrophe line. Furthermore, the A–B transition in the IC (×) occurred at a lower temperature than the transition in the HEC, consistent with the other depressurization runs terminating in this part of the phase diagram (see Fig. 3b and its inset).

## Discussion

We analyze our data by considering the model from our introduction, where the B phase grows from seeds that are contained in small chambers with a distribution of narrow necks. While cooling through the A phase, the A phase intrudes on the chambers with the largest openings: the size of the remaining channels connecting to B seeds determines the path-dependent location of the observed A–B

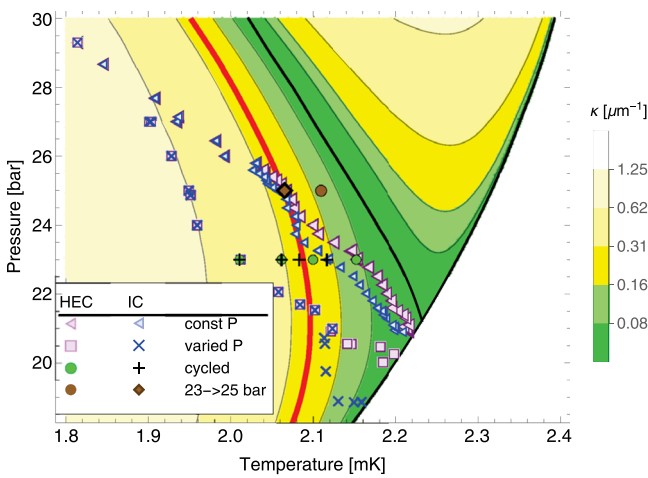

**Fig. 5 | Curvature of stable domain walls.** Contours show $\kappa_A/\sin(\theta_A)$ and $\kappa_B$ in the A and B portions of the phase diagram. The smallest curvature contour (between the two shades of green) corresponds to 0.078 µm⁻¹, while each subsequent contour is a factor of 2 larger. $\kappa$ is the mean curvature of a stable A–B domain wall, and $\theta$ is the contact angle of a domain wall with a surface. Here we assume minimal pairbreaking (specular scattering) boundary conditions. In the A phase, $\kappa_A/\sin(\theta_A)$ quantifies the inverse size of orifice which can block the motion of an A–B domain wall, while $\kappa_B$ represents the same quantity for the B phase. Under the assumption that the B phase is seeded from chambers with small openings, the largest A phase value of $\kappa_A/\sin(\theta_A)$ will set the $\kappa_B$ where the A → B transition is observed. Red line shows a contour, $\kappa = 0.25$ µm⁻¹, which roughly corresponds to where domain walls pass freely through the 1.1 µm channel between the HEC and IC, corresponding to a contact angle of 74°. To the left of this line, transitions in the two chambers always occur simultaneously.

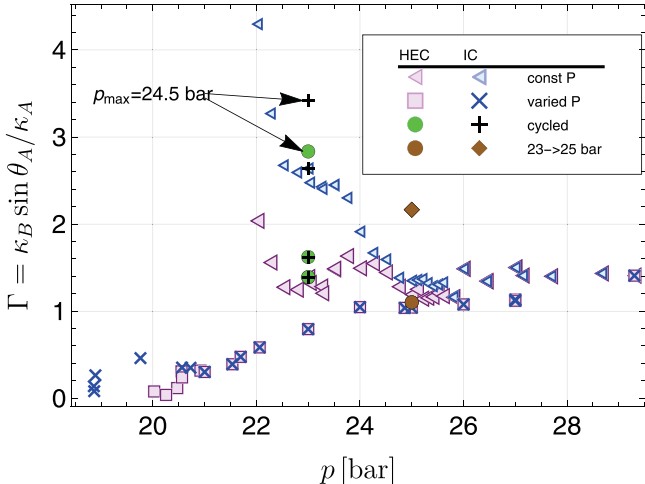

**Fig. 6 | Pressure dependence of $\Gamma = \kappa_B \sin(\theta_A)/\kappa_A$.** Ordinate, $\Gamma$ is the ratio of the domain wall curvature at catastrophe point, $\kappa_B$, to the largest scaled curvature traversed in A phase, $\kappa_A/\sin(\theta_A)$. The contact angle $\theta_A$ depends on the boundary condition: here we use minimal pairbreaking. The abscissa shows the pressure at the observed A–B transition. The collapse of the data around unity above $p = 24$ bar suggests that trapped pockets of B-phase seeds are responsible for the observed A–B transition at high pressure.

transition. Similar logic should apply to the cases where the B seeds are at the nodes of frustrated textures or distortions.

In order to balance forces, an equilibrium domain wall between the A and B phases must be bowed with a mean curvature $\kappa = |\delta f|/(2\sigma)$, where $\sigma$ is the surface tension and $\delta f$ is the difference in free energy densities between the two phases. As detailed in Supplementary Note 3, a circular hole in a flat plate with diameter $W$ will prevent the intrusion of the A phase if $W < \sin(\theta_A)/\kappa_A$, where the contact angle $\theta_A$ is determined by surface energies. Conversely, in the B region of the phase diagram, the same orifice will prevent the B phase from exiting if $W < 1/\kappa_B$—the contact angle does not appear in this expression because the B phase typically does not wet a surface. Note: these equations are sensitive to our modeling of the geometry of the orifice, and the contact angle depends on the surface properties.

In Fig. 5 we show contours of constant $\kappa_A/\sin(\theta_A)$ and constant $\kappa_B$, calculated using a Landau–Ginzburg theory and assuming minimal pairbreaking boundary conditions, corresponding to smooth surfaces. (See Supplementary Notes 2 and 3. Supplementary Note 5 deals with the results obtained for maximally pairbreaking boundary conditions. There is some ambiguity in the temperature dependence of the Landau–Ginzburg parameters, and Supplementary Note 6 discusses an alternative model.) As can be seen, $\kappa$ vanishes at the equilibrium A–B transition, where the two phases have the same free energy. It also vanishes at $T_c$. The dark green regions show where it is small. The largest values of $\kappa_A/\sin(\theta_A)$ are found at high pressure, and the largest values of $\kappa_B$ are found at low temperature. Our model would predict that for a given cooling trajectory, $\kappa_B$ at the A–B catastrophe line will coincide with the largest value of $\kappa_A/\sin(\theta_A)$ encountered while cooling: i.e., $\Gamma = \kappa_B \sin(\theta_A)/\kappa_A = 1$. For example, a constant pressure-cooled trajectory at 23.25 bar will almost touch the contour between the light and dark green regions in the A phase. The A–B transition is therefore expected to be at the same contour in the B phase. All of the varied-pressure trajectories pass through the A phase close to the contour

that separates the two lightest shades of yellow—and one therefore expects the catastrophe line to follow the corresponding B contour.

To better quantify the data, in Fig. 6 we plot the ratio $\Gamma = \kappa_B \sin(\theta_A)/\kappa_A$ vs. the pressure at which the A–B transition was observed. There is a remarkable data collapse for all pressures above 24 bar, despite the fact that the trajectories (pressure-varied or constant-pressure) are very different. The ratio is larger than the expected value of 1 (likely the result of the model's assumptions) and is essentially constant. Variations in the geometry or boundary conditions could cause this ratio to be different from unity—for example, the contact angle could be slightly less than what is predicted by the theory. Non-circular interfaces or tapered channels could skew the ratio. Analyzing the data under the assumption of maximal pairbreaking conditions further increases the ratio (see Supplementary Figs. 6 and 7). The physically relevant boundary condition lies between minimal and maximal pairbreaking conditions[49]. Previous experiments have directly tested aspects of our model of the AB phase boundary[50], including measuring equilibrium contact angles, surface tensions, and surface energies at low pressure. A number of theoretical works have also addressed the issue[9,51,52].

Below 24 bar the fixed pressure HEC and IC A–B transition data separate. Below 20.5 bar, a similar separation occurs in the pressure-varied runs. These features naturally correspond to when the channel connecting the HEC and IC can no longer support a domain wall. This feature is apparent in Fig. 5, where we draw a red line that corresponds to the contour with $\kappa = 0.25$ µm⁻¹. To the right of this line the transitions in the IC and HEC occur independently, while to the left they occur simultaneously. The IC data points which follow this red line correspond to events where the pre-existing B phase in the HEC propagates through the channel, and do not represent independent nucleation events. This includes the points below the PCP accessed by depressurization and then cooling at constant pressure. As argued in Supplementary Note 3, the B phase can propagate into the channel when $\kappa = \cos(\theta)/W$, where $W = 1.1$ µm is the height of the channel. Our inferred contact angle ($\theta \sim 75°$) is larger than typical values predicted by the Landau–Ginzburg theory with minimal ($\theta \sim 30°$) or maximal ($\theta \sim 60°$) pairbreaking boundary conditions (see Supplementary Fig. 4). This may be a feature of the glass and silicon surfaces in the channel, or it may point toward limitations in the accuracy of our theoretical model.

Between 22.5 and 23.8 bar, the constant pressure-cooled transitions in the HEC continue to agree with our model, with $\Gamma \sim 1$ (pink triangles). Over the same range the IC data clusters near $\Gamma \sim 2.5$ (blue triangles), and it is likely that the transition is completely independent of the HEC. This clustering suggests that a similar model may apply there, but with different surface geometries and boundary conditions, or the involvement of different order parameter structures. The HEC contains sintered silver, while the IC incorporates as-machined coin silver surfaces, with no obvious cavities (and channels) which could be playing the role of the B-containing seeds in the sinter in the HEC.

An additional potential mechanism for heterogeneous nucleation involves the presence of surface defects, or features, which favor a distorted order parameter. The simplest model would treat this as a B-phase seed, pinned at the surface with an associated A−B interface. The model of the catastrophe line would be analogous to the one we proposed for the sinter[53]. Given the multicomponent nature of the superfluid $^3$He order parameter (a complex $3 \times 3$ matrix), the nature of the spatially dependent order parameter of this seed region is complex. The path dependence could reflect evolution of the order parameter structures that alter the energetics of the transformation from the A phase to the B phase, without the benefit of an actual interface that would be present if a "seed" of B phase were present.

The curvature $\kappa_A$ vanishes as one approaches the polycritical point from above, and hence $\Gamma$ diverges near there for all of the constant-pressure data. At these pressures, the distribution of B seeds is likely determined by kinetic processes occurring during the normal-superfluid transition rather than details of the cooling trajectory. As emphasized in our previous work[20], it is surprising that we form the A phase when cooling at pressures below the PCP even in magnetic fields below 0.1 mT. Perhaps, since superfluidity in bulk must be induced by the colder liquid in the sinter (where the order parameter is likely distorted by surfaces), the energy cost of an interface between B in bulk and a surface-induced A phase in the sinter is too great, and the A phase is nucleated in bulk. The same scenario could follow in the IC with the channel playing the role of the sinter.

Below 24 bar, $\Gamma$ falls for the pressure-varied data. This suggests that a separate mechanism is at play: The A−B transition occurs at a higher temperature than predicted by our model. Below 20.5 bar, the transition in the IC and HEC is separate. The ratio $\Gamma$ for the HEC data continues to fall, further indicating a mechanism in the HEC which goes beyond our model. Between 20.5 and 19 bar, the transition in the IC is likely not an independent nucleation event, but rather due to the A−B domain wall breaking through the channel (corresponding to the red line in Fig. 5). This appears as a plateau Fig. 6. The cluster of IC transitions at 19 bar are likely independent nucleation events.

Figure 6 contains two additional outliers. The green discs and black crosses show $\Gamma$ for the pressure-cycled transitions depicted in Fig. 4. The trajectories that cycled to 26 bar and 27.5 bar agree very well with the rest of the data. The trajectory that cycled to 24.5 bar, however, shows more supercooling than expected, and a surprisingly large value of $\Gamma$. While we do not understand why the HEC shows such a large degree of supercooling, the IC transition coincides with the red line in Fig. 5, and is likely due to the physics of the superfluid in the channel connecting the chambers. Similarly, the brown diamond corresponds to the trajectory in Fig. 3b which was cooled at $p = 23$ bar to 2.2 mK, pressurized to 25 bar, and then further cooled. It also lies on the red line in Fig. 5 and is presumed to correspond to the B phase being conveyed through the channel. The transition in the HEC for this trajectory (brown disc) agrees well with our model.

Finally, we note that the domain wall between the A and B phases has a finite width, extending over a few temperature-dependent coherence lengths (see Eq. 5 in Supplementary Note 2 and Supplementary Fig. 2). Near $T_c$ the temperature-dependent coherence length diverges, $\xi(T) \approx \xi_{GL}(1 - T/T_c)^{-1/2}$. The resulting "thick" domain walls are likely to have different elastic properties and may not have a well-

defined curvature. This feature may account for some of the decreases in the ratio $\Gamma$ plotted in Fig. 6 for the pressure-varied runs that extended to transitions near $T_c$.

While we have developed a coherent picture, we emphasize that a number of mysteries still remain. First, we do not have a rigorous explanation for the appearance of the A phase upon cooling through $T_c$ for a range of pressures below the tricritical point. This A region was not seen upon warming, and hence does not represent a stable phase. Superficially similar results were observed in parallel ringing experiments by Kleinberg et al. at 0.5 mT[13]. The primary difference is that in our experiment the extent of supercooling of the A phase in the IC cuts off very sharply below 20.9 bar[20], while ref. 13 observed a much smoother termination. We believe that this difference implies that there is a distinct origin to the phenomenon. As illustrated by ref. 3, the stability of the A phase is very sensitive to magnetic fields, and the 0.5 mT field in ref. 13 was potentially responsible for their observations. Our field is smaller. Second, we do not have a model for the nucleation of B seeds (or their exact nature) during the transition from the normal phase into the superfluid. Third, we have yet to establish the exact form of order parameter features that generate those seeds. This is particularly true in the IC, which lacks any natural cavities.

In conclusion, we find that the supercooling of the A phase can be extended considerably by transiting through high-pressure regions where the A phase is more stable (measured by the ratio of the A−B free energy difference per coherence length to the surface tension). The path dependence observed here is remarkable, and is only possible because of the purity of $^3$He and the relatively large energy barriers between the superfluid phases. Importantly, the path dependence that we observe is not confined to the region of the PCP. Furthermore, we provide a quantitative model for much of the observed supercooling which can be ascribed to seeds of the B phase associated with structures in the sinter and possibly with surface defects. This investigation has led to an improved understanding of heterogenous nucleation, but a quantitative explanation awaits more comprehensive modeling of the "seeds" and their connecting channels to the bulk. We note that the supercooled liquid is stable at pressures as low as 18.6 bar, which can be contrasted to the lowest stable pressure for bulk $^3$He, 21.23 bar. We found that the lifetime of the metastable fluid exceeded one day at 19.8 bar. For a significant part of the phase diagram the degree of supercooling appears to be determined by the maximum value of $\kappa_A / \sin(\theta_A)$ encountered – a quantity that corresponds to the inverse size of an aperture that can support an A−B domain wall.

Despite these insights, aspects of the A−B transition remain enigmatic. In the superfluid $^3$He environment of this experiment, we have made a study of the systematics of heterogeneous nucleation by exploring a variety of trajectories in the pressure-temperature plane. We have shown that the surface energy of the A−B interface (strongly dependent on $p$ and $T$), and the contact angle with surfaces play a central role in this nucleation process. On the other hand, in previous work on superfluid $^3$He confined in a nanofluidic cavity[54] negligible supercooling was observed. The transformation from the A to the B phase involves a transit through a multi-dimensional landscape that could be hysteretic with pressure. To develop the A−B phase transition as a model for the first-order transitions in the early universe, identification of all mechanisms is essential. As we observe in our analog system, it is possible that the early universe was not homogeneous, but may have contained structures such as topological defects or primordial black holes, which could play a role in the nucleation of first-order phase transitions[55,56]. Further studies will include those of superfluid $^3$He confined in nano-structured environments, in which nucleation is studied in precisely engineered volumes, coupled to bulk liquid through "valves" which effectively isolate that volume from nucleation events in the bulk liquid and heat exchanger, and in which NMR or sound is used as a non-invasive probe. Similar structures might also be used to seed the non-equilibrium Polar phase and other phases

that are not by themselves stable or naturally occurring in bulk. In turn, analogs of these structures might provide insights into the underpinnings of transitions in the early Universe. At the very least, the elimination of the possibility of a new rapid intrinsic nucleation mechanism will put the understanding of the generation of gravitational waves on a firmer footing, and allow LISA observations to be used to constrain−or discover−new physics at the electroweak scale.

## Methods

### Fork operation
The two quartz forks were each driven at constant voltage (small enough so that no drive-dependent heating was observed) from a signal generator. A current preamplifier was used as the first stage of amplification before the received signal was sent to a lock-in amplifier. The lock-in's reference frequency was ported from the signal generator. By measuring and fitting the (complex) frequency-dependent non-resonant signal in the circuit, the (anti-symmetric) quadrature component of the received signal (after background subtraction) was used to infer the difference between the drive frequency and the resonant frequency, while the in-phase component was used to infer the "$Q$" or Quality factor of the fork. The forks were maintained within 10 Hz of the resonant frequency (near 32 kHz) with $Q$ factors varying from ≈40 at $T_c$, to about 200 at the lowest temperatures at high pressure. In operation, the forks could track the $Q$ well without attention. At the A–B transition, Fig. 4, the $Q$ increased by ≈10 abruptly, providing a clear signature of the transition.

### Thermometry
The temperature of the HEC detected at $T_c$ was found to lag the temperature of the melting curve thermometer (mounted on the demagnetization stage) by only 1–2 µK providing the warming and cooling rates were less than 10 µK/h. We estimate the accuracy of our inferred A–B transition temperatures to be ±3 µK, as long as the cooling rate was held constant in a given A–B transition run. The cooling rate of the nuclear stage was controlled by adjusting the rate of decrease of the current in the magnet and could be reliably set to be a constant 10 µK/h or even held constant (±3 µK) for periods as long as a day. The temperature of the fork in the IC lagged that of the HEC by ≈15 µK (inferred by observing the differences in the observed $T_c$ while cooling at constant temperature). In all graphs, the data have been adjusted for this lag. The inferred temperature of $^3$He in the IC were similarly adjusted appropriately while cooling at constant pressure if the supercooled transitions in the IC and HEC occurred at different times (i.e., below 24.5 bar see Fig. 3a).

### Pressure control
The pressure was regulated using a temperature-controlled "bomb" consisting of a ≈10 cm$^3$ volume in the form of a 9.5-mm diameter stainless steel tube. An insulated Nichrome wire was wound on this tube and was connected to a 0–25 W power source whose output was set by a digital proportional integral and differential controller. The bomb was semi-isolated from the lab environment by being contained in a large cylindrical tube 5 cm id × 25 cm long, open at both ends and mounted vertically. The pressure could be monitored by a digital Heise DXD 0-40 bar pressure gauge, allowing for high resolution with minimum volume in the system. A 0.3 cm$^3$ volume filled with silver sinter was used as an additional heat sink to thermalize the $^3$He before it entered into the main HEC chamber. In this way, we were able to vary the pressure by as much as 5 bar/day without incurring significant heating, allowing large pressure changes to be effected quite rapidly. A–B transitions were observed while the pressure was constant or varied by 0.7–1.3 bar/day in order to avoid any issues with viscous heating in the channel. Due to the limited heat, we could apply to the bomb, pressure changes greater than 5 bar required multiple steps, where we would stop and reset the pressure in the external gas handling system.

## Data availability
The $P$, $T$ data generated in this study have been deposited in the Cornell University e-commons data repository database under accession code https://doi.org/10.7298/1sw8-f758.

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

## Acknowledgements

This work was supported at Cornell by the NSF under DMR-2002692 (J.M.P.), PHY-2110250 (E.M.), and in the UK by EPSRC under EP/J022004/1 and by STFC under ST/T00682X/1 (M.H., J.S., K.Z.). In addition, the research leading to these results has received funding from the European Union's Horizon 2020 Research and Innovation Programme, under Grant Agreement no 824109 (J.S.). Fabrication was carried out at the Cornell Nanoscale Science and Technology Facility (CNF) with assistance and advice from technical staff. The CNF is a member of the National Nanotechnology Coordinated Infrastructure (NNCI), which is supported by the National Science Foundation (Grant NNCI-1542081).

## Author contributions

Experimental work and analysis was principally carried out by Y.T. with early contributions by D.L. and A.E. assisted by A.C. with further support from E.N.S. and J.M.P. Presentation of figures was the joint work of Y.T. and A.E. assisted by K.Z., M.H. and D.L.. N.Z. had established most of the routines for the phase locked loop operation of the quartz fork for earlier experiments. E.M. significantly contributed to exploration of the phase diagram and the writing of the manuscript, and N.Z. and T.S.A. established and carried out the nano-fabrication of the channel. M.H. and K.Z. in conjunction with E.M. explored the relationship of $\kappa$ to $\sigma$ and calculated the contours of constant $\kappa$. J.M.P. supervised the work and J.M.P., E.M. and J.S. had leading roles in formulating the research and writing this paper. All authors contributed to revisions to the paper.

## Competing interests

The authors declare no competing interests.
