## [Peer Review File · Nature Communications]

REVIEWER COMMENTS

Reviewer #1 (Remarks to the Author):

The manuscript “Supercooling of the A phase of ^3He ”, by Y. Tian et al. describes a careful investigation of the supercooling of the A-phase of ^3He . It is essentially the logical continuation of a systematic study (technical details and some physical results have already been published (Nature Communications 11, 4853, 2020 and Phys. Rev. Lett. 126, 215301, 2021).

The path-dependent supercooling effect discussed in the present manuscript was already observed in these previous works, but in a very limited range within the p-T phase diagram, in the vicinity of the polycritical point (coexistence of normal fluid and A and B superfluid phases). In the present work, a wide range of pressures is investigated, using different thermodynamic paths to cross the A-B first order transition line. A considerable amount of information has been obtained. The experiments are difficult and time consuming, they have clearly been made with great care and dedication.

The analysis of the results is sound, considering reasonable scenarios able to explain the heterogeneous nucleation of the B phase, observed in two different conditions: in an Isolated Cell (IC) and in the Heat Exchanger Cell (HEC), connected by a thin channel. The HEC contains sintered silver powder, which clearly plays a major role in the B-phase nucleation, as argued in this manuscript, by providing a “seed-reservoir” for the transition. Different temperature regions are identified, associated to the thin-channel ability to stop the propagation of the A-phase. Theoretical arguments and calculations are given, qualitatively but convincingly supporting the proposed “seed model”.

The authors also list several effects, observed here but not explained by this model, in particular around the polycritical point. The nature of the “cavities” and the evolution of the “seeds can be investigated in more detail, and plans for future work are briefly discussed.

The results are clearly of interest to the low temperature community, and I sincerely enjoyed reading this manuscript. They are also of interest, in principle, to Cosmologists who investigate phase transitions in the Early Universe, as is clearly explained in the manuscript, with examples. Even if a very detailed analogy fails, ^3He it is the closest laboratory approximation, and certainly a good source of ideas and motivation for both fields of research.

The present work does obviously not “solve the problem” of heterogeneous nucleation. This not surprising, and the authors make a very clear statement at this respect. The free energy landscape as a function of the intensive parameters [p,T] is already very complicated in bulk ^3He . Needless to say, adding the effect of the surfaces, textures, distortions (it is not clear if the authors consider vortices among these...), which can modify significantly the free energy landscape, makes the problem extremely awkward. The authors make a clear effort in order to describe theoretically some simple cases, with a rather convincing success (Fig. 5 and 6) supporting the suggestion that trapped pockets of B phase are responsible for the A-B transition observed at high pressure.

The experiments are conducted at zero (Earth) magnetic field, indeed negligible in the present case, but I am nevertheless surprised that the well-known effect of this parameter around the polycritical point (Paulson, D. N., Kojima, H. & Wheatley), is not introduced in the discussion: a magnetic field strongly favors the A phase in this region, and surfaces too, a qualitative and perhaps even a quantitative comparison could have been made.

To conclude about the relevance of this work, I clearly agree with the authors' claim that this work provides significantly more clarity about the phenomenon. The subject is very complex, but of genuine general interest.

I would like to make a few comments about the manuscript itself. I have to admit that when I first read it, I thought that important information about previous experiments on nucleation in ^3He was missing in the introduction. In fact, the details are given further down in the manuscript, and this is, I believe, very clever: the reader can follow the main lines of the manuscript without being disturbed by technicalities of superfluid ^3He , complexities are deferred to the moment where they are really needed. The hard core of the discussion is not easy to swallow, an unavoidable feature, given the complexity of the system and the large number of independent experiments ("thermodynamical paths") realized here. The whole text (including the Supplemental Material) is detailed, rigorous, and particularly clear. I appreciate the fact that results are reported honestly, pointing out potential difficulties in the experimental set-up, limitations of the proposed models, and in the suggested interpretation of the results.

My overall conclusion is that this manuscript constitutes a significant advance in the investigation of heterogeneous nucleation, and I recommend its publication in Nature Communications in its present form.

Suggested improvements:

"200 μm tall \times 3 mm wide \times 2.5 mm long lead-in channels on either side". "Wide" should be replaced by "diameter", to avoid confusions with the Letterbox dimensions mentioned just before. These cylindrical channels could be identified (arrows) in Figure 2, and a circle could be drawn around the 1.1micrometer channel, at the tip of the corresponding arrow.

Misprints: "Propogating" "propogates" could be properly corrected

Reviewer #2 (Remarks to the Author):

The manuscript by Tian Y. et al. "Supercooling of the A phase of ^3He " describes new measurements of supercooling ^3He -A in a wide range of parameters. The manuscript presents a comprehensive attempt to understand the first order transition in the chemically cleanest environment available on

Earth. The experimental results are sound and clearly show that temperature and pressure path dependencies influence the transition from A to B phase (degree of supercooling).

The authors propose a theoretical model linking pockets of B phase that are formed in the wall cavities after the superfluid transition to a degree of observed supercooling. Figures show a clear link between the model and various path trajectories measured. The model works extremely well above 25 bars where the paths data collapses but needs further development/thoughts at lower pressures (which the authors plentifully provide). The future work is carefully considered. Overall, the manuscript is well written and warrants publication in Nature Communications. It significantly expands boundaries of our understanding (or lack of it) of the first order transition and is relevant to a wider audience as it has a direct link to physics of the early universe.

There are a few minor comments, which should be considered.

Figure 1 colors are not color-blind friendly with a mix of red and green lines. The inset symbols referring to the [19] are not described in the manuscript and the magnetic fields are not stated.

Figure 2 could be made clearer (3D) in the body of manuscript and in the supporting information. The description of where the 1.1 micron channel is and how it relates to the 200 micron channel should be shown in the figure, which offers an ample space to do so.

Reviewer #3 (Remarks to the Author):

This paper studies the path dependence of the supercooling of the A phase of He-3 over a wide range of pressures. In particular, the authors investigate the nucleation of the B phase of He-3 by using two chambers filled with bulk He-3 that allow observation of the A→B transition using quartz fork resonators. The experimental results are quite wonderful. For example, Fig. 3 shows the paths in (P, T) space for constant pressure and pressure-varied phase transitions and is a great illustration of the intricacy of first-order phase transitions.

One particular insightful result is that there appears to be more supercooling in Fig. 3(b) (pressure-varied transitions) than in Fig. 3(a) (constant pressure). Supplementary-Note 4 provides a helpful guide to show that the enhanced supercooling in Fig. 3(b) is unlikely to be due to systematic effects. The results at large pressures, $P > 24$ bar, have a constant Γ quite close to the authors' prediction of $\Gamma = 1$, which gives credence to their model. The scientific analysis is careful and sound – the strong-coupling GL analysis in the Supplement and the determination of Γ are valuable additions to the manuscript.

However, I am unconvinced that this paper is appropriate for Nature communications. Indeed, in the May issue of PRL last year, the authors already presented an initial study of path-dependent supercooling in the A-B transition. Figure 4 of that paper showed similar results to those found here, namely the paths in (P, T) space for A→B transitions. Granted the present study occurs at greater pressures (29.3 bar as opposed to 22 bar), but the presence of enhanced supercooling already appeared in a prestigious journal. The Γ analysis and the semi-reasonable agreement with the authors' prediction of $\Gamma = 1$ is indeed a new result found at these larger pressures; nevertheless, much more experimental and theoretical analysis at large pressures needs to be done.

While the experimental results are interesting, and there is some analysis beyond the authors' 2021 PRL, I do not think there is a sufficient enough leap beyond the PRL paper to warrant publication in Nature communications.

A few minor comments for the authors to expand or reflect upon are as follows:

(i) What fundamental new ideas are contained in this manuscript that could not be understood from the 2021 PRL by the same authors?

(ii) In several places the authors allude to the possibility for He-3 to describe phase transitions in the early-universe. He-3 is indeed a rich system, however, if the authors intend for a meaningful comparison between their results of phase transitions in the A-B phases of He-3 and cosmology models, then they must give further details. Citing Volovik's book is insufficient.

For instance, He-3 has vortices and defects, which are expected to mimic cosmic strings, does the present study give any further understanding about defects and vortices in He-3? Are there any new insights related to the Kibble-Zurek mechanism etc.?

(iii) In regard to the authors remarks on page 3 concerning models of nucleation theory in the early universe. Again, more detailed exposition is required. Coleman and Callan studied a relativistic quantum field theory where the false vacuum is a metastable state of the scalar field. Helium-3 is not the same entity. There have been other condensed-matter proposals for testing Coleman and Callan's idea in the context of ultra cold atoms [EPL, 110 (2015) 56001, JHEP 07 (2018) 014]; therefore, one should not overestimate the relevance of He-3 to such interpretations without more detailed analysis. As the authors note on page 6, thermal fluctuations can also influence nucleation, and the paths connecting A and B phases in He-3 can be strongly influenced by confinement effects.

The He-3 results are interesting enough on their own without conflating them with possibly related (or unrelated) physics.

(iv) The values of Γ in Fig. 6 are closer to ~ 1.4 than to unity. Are there any geometrical or physical reasons for this noticeable discrepancy?

Dear Editor and Reviewers

The authors would like to thank the reviewers for their thorough reading of our manuscript, and for the many substantive comments that we have used to recast some portions of the manuscript. We believe that these changes both strengthen and add to the clarity of the presentation.

We have included in the materials submitted, a pdf copy of the paper with substantive changes to the text highlighted in blue. No substantive changes were made to the Supplemental Materials.

Below we address remarks to each of the Reviewers with reference to page numbers and paragraph numbers. We paraphrase specific paragraphs and then add the remarks below.

Reviewer #1

We thank the reviewer for his many compliments and summary.

1. The experiments are conducted at zero (Earth) magnetic field, indeed negligible in the present case, but I am nevertheless surprised that the well-known effect of this parameter around the polycritical point (Paulson, D. N., Kojima, H. & Wheatley), is not introduced in the discussion: a magnetic field strongly favors the A phase in this region, and surfaces too, a qualitative and perhaps even a quantitative comparison could have been made.

We now include (p16-17) a brief discussion about the low field (0.5mT) results from Wheatley's group (Ref12) where the A phase was seen on cooling below the Polycritical point, but did not appear on warming. (see also a possible scenario for A phase nucleation in the bulk below the PCP on p15). The higher field results that the reviewer was pointing us toward show the A phase reappearing on warming (the so called Profound Effect) (Ref 3).

2. Suggested improvements:

“200 μm tall \times 3 mm wide \times 2.5 mm long lead-in channels on either side”. “Wide” should be replaced by “diameter”, to avoid confusions with the Letterbox dimensions mentioned just before. These cylindrical channels could be identified (arrows) in Figure 2, and a circle could be drawn around the 1.1micrometer channel, at the tip of the corresponding arrow.

Figure 2 has been redrawn following the reviewer's suggestions. In fact the “lead in” channels are rectangular, while the connecting tube is circular. These have been clarified in the revised figure and caption (p31).

3. Misprints: “Propogating” “propogates” could be properly corrected
Fixed.

Reviewer #2

We thank the reviewer for his thorough reading of the manuscript and his understanding of the problem described in the manuscript.

There are a few minor comments, which should be considered.

1. Figure 1 colors are not color-blind friendly with a mix of red and green lines. The inset symbols referring to the [19] are not described in the manuscript and the magnetic fields are not stated.

We have modified the colors in the figure and added text to the caption addressing the field magnitude and identified the symbols. (p30)

2. Figure 2 could be made clearer (3D) in the body of manuscript and in the supporting information. The description of where the 1.1 micron channel is and how it relates to the 200 micron channel should be shown in the figure, which offers an ample space to do so.

Figure 2 has been redrawn following the reviewer's suggestions and those of Reviewer #1. (p31)

Reviewer #3

We thank the reviewer for his thorough reading of the manuscript and his many comments regarding the manuscript and supplement.

We understand the reviewer's concerns regarding the issue of this more comprehensive study and its potential overlap with the PRL published by a subset of the authors last year. As the reviewer already posed a set of cogent questions to us, we phrase our response in the form of replies to those questions, together with modification to the text of the paper.

1. What fundamental new ideas are contained in this manuscript that could not be understood from the 2021 PRL by the same authors?

We showed that this phenomenon is not limited to the region near the PCP.

Importantly, we provide a quantitative model that shows that the maximum curvature of the A-B interface attained along a constant pressure or pressure varied path in the equilibrium A phase is linked to the location of the A to B supercooled transition. We further attribute this to seeding presumably associated with the sinter.

This is thus the first quantitative explanation for the nucleation of the B phase from the supercooled A phase aside from those studies where external means such as energetic particles were introduced.

Thus, a mechanism for heterogenous nucleation has been identified.

These features are now highlighted on p 17 in the concluding paragraphs of the paper.

2. In several places the authors allude to the possibility for He-3 to describe phase transitions in the early-universe. He-3 is indeed a rich system, however, if the authors intend for a meaningful comparison between their results of phase transitions in the A-B phases of He-3 and cosmology models, then they must give further details. Citing Volovik's book is insufficient.

This is an important point, and we realize that it warrants further development. We addressed this by adding text and references on page 3, and further on Page 5 addressing the Kibble-Zurek mechanism. (see the next point as well).

3. For instance, He-3 has vortices and defects, which are expected to mimic cosmic strings, does the present study give any further understanding about defects and vortices in He-3? Are there any new insights related to the Kibble-Zurek mechanism etc.?

We now include a statement that the Kibble-Zurek mechanism is unlikely to be involved on account of the cooling rate independence of the extent of supercooling (albeit at low pressure where the pulsed experiments described in Ref 19 were performed). We also acknowledge that we lack the thermometry to perform calorimetric measurements (Ref 38) that were used to infer the probable existence of a vortex tangle. (see page 5).

4. In regard to the authors remarks on page 3 concerning models of nucleation theory in the early

universe. Again, more detailed exposition is required. Coleman and Callan studied a relativistic quantum field theory where the false vacuum is a metastable state of the scalar field. Helium-3 is not the same entity. There have been other condensed-matter proposals for testing Coleman and Callan's idea in the context of ultra cold atoms [EPL, 110 (2015) 56001, JHEP 07 (2018) 014]; therefore, one should not overestimate the relevance of He-3 to such interpretations without more detailed analysis. As the authors note on page 6, thermal fluctuations can also influence nucleation, and the paths connecting A and B phases in He-3 can be strongly influenced by confinement effects. The He-3 results are interesting enough on their own without conflating them with possibly related (or unrelated) physics.

We have expanded our discussion concerning the connection to nucleation theory in the early Universe and ^3He , and reworded it to make clearer the role of thermal fluctuations, although quantum fluctuations (as per Callan and Coleman) have also been considered (and dismissed) as a possible mechanism. This background information has been transferred to p5 where we discuss alternative explanations. We thank the referee for the ultracold atom references that have been added [37,38] to the MS on p3. We agree with the referee that quantum tunnelling is not relevant to our experiment, and hope that this is now clearer.

5. The values of Γ in Fig. 6 are closer to ~ 1.4 than to unity. Are there any geometrical or physical reasons for this noticeable discrepancy?

The reviewer is correct. The modelling of realistic shapes of connecting pathways in the sinter is clearly something beyond the scope of this paper. However, one can show that a rectangular channel as opposed to a cylindrical channel, would yield a larger ratio, as is certainly the case for a tapered channel. We believe that the cylindrical channel likely represents a lower bound for this ratio. We now include a statement on page 13. The ratio is also affected by the choice of boundary conditions (compare figure 6 in the paper to Figure 7 in the supplemental using maximal pairbreaking conditions). The actual configuration of the order parameter in the “seeds” is beyond the scope of this paper.

Jeevak Parpia
For Authors